



# Variable Saturation Infiltration Model for Highly Vegetated Regions

James Polsinelli[1] and M. Levent Kavvas[1]

[1]Department of Civil Engineering, University of California-Davis, Davis, California

*Correspondence to:* James Polsinelli (jfpolsinelli@ucdavis.edu)

**Abstract.** Observations have been made in studies that in watersheds with moist soils and lush vegetation, water does not pond on the soil surface, even during significant rainfall events. In these cases, all the water infiltrates into the soil. Furthermore, the water content in the pore space is below saturation.

The tools for modeling such situations are limited. A universal method for estimating groundwater in the unsaturated zone
is the Richards' equation, a non linear system of 1D or 3D equations based on physical flow principles. While effective, the difficulties in solving Richards' equation can pose a significant problem in a complex system, which may require extensive amounts of calibration and numerical effort in procuring a solution. A second method is based on the idea of approximating the movement of moisture through the soil as a rectangular profile, e.g. assuming that the water moves through a soil column like a piston. Rectangular profile methods have been developed in the past for cases in which ponding occurs at the surface, and
the water content inside the column is at saturation. More general rectangular profile methods allow the water in the column to move at sub-saturation. Both of these existing methods involve solving a non linear algebraic equation for the hydrologic system, involving constant flux at the boundary, due to a wetting event such as rainfall. The method proposed in this article may be used for situations in which the soil does not become saturated, and the soil and rainfall properties, specifically the rates of hydraulic conductance and rate of rainfall respectively, are allowed to vary in space or time. The method proposed is based
on a simple principle in infiltration hydrology, that the rate at which water will infiltrate through the soil equilibrates with the rate of rainfall, when the rate of rainfall is smaller than the infiltration capacity of the soil. In application, this method does not require the solution of a non linear algebraic (or differential) system of equations, thus affording modelers computational economy.

Furthermore, this new method can be made to interact with the saturated profile methods in the event that the rainfall
overwhelms the ability of soil to absorb moisture. If this happens, a portion of the field will come to saturation while other parts will be below saturation. The sub-saturation models can be made to interact with the redistribution and evapotranspiration processes by providing the initial and boundary condition for those events.

## 1 Introduction

Infiltration through the unsaturated zone in the subsurface is complicated by a number of factors. Early experimental work
conducted by Darcy in the 1850's concluded with an empirical relationship between the flux and the hydraulic gradient for one dimensional flows. This relationship came to be known as Darcy's law and is the basis for most of the groundwater studies





since. A thorough theoretical treatment was performed in Whitaker (1986) to prove the 3D tensorial analogue of Darcy's law, and to supply both the 1D and 3D Darcy relationships with a solid physical and mathematical basis. Furthermore, the theoretical derivation of Darcy's law by *Whitaker* does not require the assumption of a spatially periodic or homogeneous medium. The two main techniques for representing groundwater flow in the unsaturated zone are based on the 1D and 3D Darcy's law, respectively.

Representation of heterogeneity in soils is a difficult problem by itself. Expressing the variability in soil properties is important in any description of subsurface dynamics. Many investigators use a stochastic description of variability in the porous matrix. See Freeze (1975); Philip (1980); Dagan and Bresler (1983) for examples. Heterogeneity is usually accounted for by variation in the saturated hydraulic conductivity, $K_s$, a lumped parameter of both fluid and medium properties. Stochastic models vary, but it is most usual for $K_s$ to be taken as a random variable having a log normal distribution. Other models assume that $K_s$ is a random process.

An established set of governing equations, known as the Richards' equations since Richards (1931), describe the subsurface distribution of water content and head through conservation of mass combined with Darcy's law. Richards' equations can be developed for 1D or 3D, and are categorized as a system of nonlinear advection dispersion type equations. The non linearity of these equations make them difficult to work with, even in homogeneous soils, though a significant amount of work has been done to develop suitable solutions. See Celia et al. (1990) for a thorough description. Using the Richards' equations in either 1D or 3D is further complicated when the soil is non uniform. If a stochastic representation is used to account for soil/flow variability, then the non linear Richards' equation becomes a non linear stochastic partial differential equation. In this case the system is difficult to solve even numerically, because of the immense number of parameters that must be calibrated in a grid with sufficient resolution to be useful in hydrological applications. In these cases, researchers more often turn to rectangular profile approximations for subsurface modeling.

Another popular model used to represent subsurface flows is the rectangular profile model, first introduced in Green and Ampt (1911). This model uses Darcy's law and the assumption that moisture moves through the soil in a piston-like fashion. The spatial distribution of water in the soil is taken as having a constant water content (e.g. constant ratio of water volume to void volume in a given pore volume) above a sharp front, and initial water content below the front. Generally, in sharp front models, the water content above the front is assumed to be at saturation. Hence, water moves through the soil by filling up all the pore space in a wetted volume cell before proceeding to deeper cells. This assumption simplifies the non linear hydraulic conductivity function to a constant value of the conductivity at saturation, known as the saturated hydraulic conductivity. These models are usually used in cases where the infiltration rate will exceed the ability of the soil to conduct water, and water will pond on the surface of the soil exposed to the atmosphere. Within this framework the time to ponding can be explicitly found. It is assumed that prior to the time to ponding, all the water infiltrates into the soil. After ponding, the infiltration depth can be found using the Darcy's law, and the assumption that the water content above the front is at saturation. It turns out that after ponding, the infiltration rate into the soil decreases non linearly. The problem can be shown to take the form of a nonlinear algebraic equation that is generally solved either by inspection or iteration Rawls et al. (1983). The depth of the wetting front can be found at any desired time between the time to ponding and the cessation of the wetting event allowing a closed form





representation of all hydrologic quantities in the domain of interest. The analysis that follows will assume a semi-infinite unsaturated column. This is to say that the water table is not considered as a boundary condition in this study.

The new method proposed in this paper is based on a rectangular profile assumption similar to the Green and Ampt method. An assumption is made that if the rainfall rate is less than the infiltration capacity of the soil, then the hydraulic conductivity

immediately equilibrates with the rainfall rate. This is to say that the hydraulic conductivity is immediately equal to the rainfall rate when the rainfall rate is smaller than the saturated hydraulic conductivity. This phenomenon has been observed in humid regions with dense vegetation, and has been documented in (Dunne and Kirkby, 1978, Dunne, 1978). Since the hydraulic conductivity is a function of the water content, there is a specific water content value associated with the hydraulic conductivity being equal to the rainfall rate. This special water content will be the water content in the column of water infiltrated into the

porous matrix above the sharp wetting front. In the case that the rainfall rate exceeds the saturated hydraulic conductivity, the model reverts to the classic saturated rectangular profile known as the Green and Ampt model.

There are other models as well, some empirical and some physically based. In this paper, the focus will be on the Green and Ampt model as the classical infiltration model, because of its widespread use, and a preference towards physically-based models. References for other models can be found in Charbeneau (2000).

## 15    2    Classical Infiltration Modeling

Determination of infiltration from rainfall events into the unsaturated zone in the subsurface is usually studied by using the Green and Ampt model. This model is based on laboratory and field observations that water moves through a soil column like a piston. This movement is characterized by the presence of a sharp front. Above the front the soil is completely saturated, and below the front the water content is what it was before the rainfall event. At the cessation of the rainfall event, a drying front is

20 born at the soil surface and proceeds downward. The eventual fate of the rainwater is to reach the water table, or be absorbed by other processes such as evapotranspiration, etc.

A critical assumption made by the Green and Ampt model is that ponding at the surface occurs at some time before the cessation of rainfall. After ponding, a relatively simple differential equation may be constructed by taking a depth integral of the flow equations up to the location of the wetting front Rawls et al. (1983). An equation relating the infiltration capacity rate

to the cumulative infiltration can be formulated from the depth integrated flow equations. The infiltration capacity rate is the time derivative of the cumulative infiltration in a column of soil, and the cumulative infiltration can be related to the wetting front depth through the rectangular soil moisture profile assumption. This transforms the equations into a differential equation for the location of the wetting front and the rectangular soil moisture profile relation between the cumulative infiltration and the wetting front depth. The wetting front depth is allowed to vary as a function of time in the case of a homogeneous soil. The

differential equation may be integrated exactly to obtain a closed form expression for the depth to the wetting front.

The Green and Ampt model is popular because it is easily solved, and it forms a good approximation to the dynamics in cases where ponding occurs on the surface during a rain event and the soil is mostly uniform. The utility of this exact solution cannot be overstated. It should be noted that the solution to the differential equation is given as an implicit equation for the





cumulative infiltration, which must be solved by iterative or inspectional/graphical techniques. The substantial economy of the sharp front model makes it a compelling model whenever it is usable. In situations where these assumptions do not hold, the more computationally intensive Richards' model must be used.

It is known that the hydraulic conductivity depends on both the spatial coordinates and the value of the water content (or head) at a location and time. The precise functional form of the conductivity is unknown, but its empirical models exist. A few popular models are Brooks and Corey (1964), van Genuchten (1980), and Kosugi (1996). For a summary see Simunek et al. (2009). The empirical models assume that the variation of the conductivity is a non linear function of the water content, e.g. the relative conductivity is assumed to be a power law of the saturation. The models separate the conductivity into a spatially variable saturated hydraulic conductivity [Length per time], and a dimensionless relative hydraulic conductivity. Spatial dependence is present in the variation of the saturated hydraulic conductivity as mentioned earlier. It is popular to model $K_s$ using a stationary process with the finite dimensional distributions of $\log K_s$ having a normal distribution with known mean vector $\mu$ and covariance matrix $\Sigma$: $\ln K_s(\mathbf{x}) \sim \mathcal{N}(\mu, \Sigma)$. Certain applications may demand a non stationary representation for $K_s$.

## 3 Variable saturation subsurface model for vegetated regions

A rectangular profile approximation was introduced in Chen et al. (1994a, b) which accounted for random variations in the saturated hydraulic conductivity, and allowed for rectangular profiles with water content below saturation. Examples were given for two cases: constant saturation at the soil surface, and constant flux at the soil surface. In the former case, the saturation throughout the rectangular profile is the same as the surface, so only the depth to the wetting front is unknown. In the latter case, both the saturation at the surface and the depth to the wetting front are unknown. A number of bifurcations were identified in the values of the saturated hydraulic conductivity and the depth. These bifurcations sorted out where the soil was saturated, where it was below saturation but still in the wetted column, and where it was at initial water content (beyond the reach of the wetting front). Care was taken to express the saturation at location $z$ at time $t$ in terms of the value of the saturated hydraulic conductivity $K_s(\mathbf{x})$ at the horizontal location $\mathbf{x}$. This facilitated relating the horizontally averaged saturation in terms of the expected value of $K_s$ (recall $K_s$ is a random process). I.e. for saturation $s$, horizontal area $A$, and assuming $K_s$ is a random variable with probability density function $f_{K_s}$ (e.g. lognormal) the ensemble average soil saturation $s$ at depth $z$ at time $t$ is expressed as :

$$\langle s \rangle (z,t) = \frac{1}{A} \int_A s(z,t;K_s(\mathbf{x}))\, d\mathbf{x} = \int_0^\infty s(z,t;K_s) f_{K_s}(K_s) dK_s.$$

The procedure in this study can be similarly developed, though it is less technically complicated for the flux case, since the water content and depth to the wetting front are easily determinable. For many cases, portions of the field will be saturated wherever the saturated hydraulic conductivity is less than the rainfall rate. This is bound to happen in many problems, especially in arid regions with sparse vegetation, since both the soil conductivity and the rainfall rate can vary with large magnitudes.





## 3.1 Infiltration when $i < K_s$

Consider the case when the local rainfall rate, $i$, is less than the soil saturated hydraulic conductivity, $K_s$. It will be supposed that the hydraulic conductivity immediately equilibrates with the rainfall rate. In this model there are two unknowns that must be determined and a number of statistical and empirical parameters that must be calibrated. The unknowns are the water content inside the rectangular profile and the depth to the wetting front. In order to determine the unknowns, the Darcy's law is used.

$$q_z(t) = -K(\theta_t)\frac{dh}{dz} = -K(\theta_t)\left(\frac{\partial \phi}{\partial z} - 1\right) = i. \tag{1}$$

The observations in Dunne and Kirkby (1978) indicate that $\partial \theta(K = i)/\partial z = 0$, hence $\partial \phi(\theta = \theta(K = i))/\partial z = 0$ and

$$K(\theta_t) = K_s(\mathbf{x})K_r(\theta_t) = i. \tag{2}$$

Utilizing a sharp front model for the water movement in the subsurface with water content in the front equal to $\theta_t \leq \theta_s$, it will be assumed that the cumulative specific volume of water in the subsurface domain is equal to the infiltration rate times time (or integrated over time).

$$F(t) = it = L_f(\theta_t - \theta_i), \tag{3}$$

where $L_f$ is the depth of the wetting front at time $t$.

The water content at time $t$ is found by inverting the relative hydraulic conductivity function in equation (2):

$$\theta_t = K_r^{-1}\left(\frac{i}{K_s(\mathbf{x})}\right) \ , \ \ 0 \leq t \leq t_r. \tag{4}$$

Using the Brooks and Corey (1964) power law model for relative conductivity the soil water content at $(\mathbf{x_o}, t)$ will be:

$$K_r(\theta_t) = \left(\frac{\theta_t - \theta_o}{\theta_s - \theta_o}\right)^{\frac{2}{\eta}+l+2} \ \text{ or } \ \theta_t = (\theta_s - \theta_o)(i(\mathbf{x_o}, t)/K_s(\mathbf{x_o}))^{\frac{1}{2/\eta+l+2}} + \theta_o. \tag{5}$$

The depth to the wetting front is easily found because the cumulative volume of water that has infiltrated into the soil is known. Under the rectangular profile approximation:

$$L_f(t; K_s(\mathbf{x_o})) = \frac{V}{\theta_t - \theta_i} = \frac{F(t)}{(\theta_s - \theta_o)(i(\mathbf{x_o}, t)/K_s(\mathbf{x_o}))^{\frac{1}{2/\eta+l+2}} + \theta_o - \theta_i}. \tag{6}$$

For a simple example of this variable saturation model, a numerical experiment was done comparing the model to the solution of the Richards' equation provided by HYDRUS 1D Simunek et al. (2009). The simulation was done for an infiltration problem through a uniform porous matrix consisting of sand (saturated hydraulic conductivity of 0.00583333 cm/sec), subject to atmospheric boundary conditions of constant rainfall of 0.003 cm/s for the duration of 36,000 seconds. Free drainage conditions were taken at the lower boundary. Ponding does not occur at the surface since the rainfall rate is less than $K_s$. A constant water content of 0.2001 is taken throughout the porous matrix below the wetting event.

Figure 1 demonstrates the relative performance of the model with respect to the solution to the exact 1D equation. Figure ?? displays the error between the exact and approximate solutions. The error is predictably high near the curvilinear edge of the wetting front, and low prior to the front. The relative error in the rectangular profile away from the front is under 1.5%.





### 3.2 Infiltration for $i \geq K_s$

In the application of this method, a few things must be monitored. The domain is broken up into a discrete set of grid spaces. If, for some period of time, $t \in [t_o, t_1]$, the rainfall rate in the grid cell $i(\mathbf{x_o}, t)$ exceeds the saturated hydraulic conductivity $K_s(\mathbf{x_o})$, the water content in that cell may be saturated. If $t_1 - t_o$ is greater than the time to ponding, a Green and Ampt type solution will be needed to find the depth to the wetting front. Meanwhile, if $t_1 - t_o$ is less than the time to ponding, the surface is still unsaturated.

The steps in this case follow the development of the Green and Ampt model. Ponding will occur if the time to ponding, $t_p$ is in $t_o \leq t_p \leq t_1$. The time to ponding is found through the relation:

$$t_p = \frac{(\theta_s - \theta_i)\Psi_f}{i(1 - i/K_s)}. \tag{7}$$

In the above equation, $\Psi_f$ is the suction pressure head at the wetting front. Note that the initial water content must be viewed as the water content at the time when the rainfall rate first exceeds the saturated hydraulic conductivity. Prior to the time to ponding, it is assumed that all the water infiltrates into the soil. Essentially it is assumed that the hydraulic conductance of the soil is equal to the rainfall rate for prior times to the time of ponding. The profile of moisture within the soil matrix at the time of ponding will be rectangular with a saturated moisture content, and the depth to the front of the profile will be determined based on the total volume of water infiltrated into the soil until the ponding time. Immediately after the ponding time, the infiltration rate will decrease, and the hydraulic conductance of the soil will be the saturated hydraulic conductivity. The infiltration rate is found by solving a nonlinear algebraic equation, as seen, for example, in Charbeneau (2000).

In the event that the time to ponding is not reached, e.g. the rainfall rate is below the saturated hydraulic conductivity before the ponding time, the model will transition back to the variably saturated rectangular profile model. The transition may be accomplished through a linear model, such as rectangular profile, or through non linear dynamical equations such as Richards' equation, or the kinematic wave equation.

### 3.3 The complete model prior to the cessation of rainfall

The soil saturation $s(z, t; K_s)$ will be given in terms of regimes corresponding to values of $(z, K_s)$ for a partition of the $z$ values and realizations of the random $K_s$ values. For a given value of $L = L_o$, consider the corresponding value of $K_s$ from (6):

$$K_s = K_{L_o} = i\left(\frac{\theta_s - \theta_o}{F/L_o + \theta_i - \theta_o}\right)^{2/\eta + l + 2}. \tag{8}$$

For any particular value of $L_o > 0$ if $K_s > K_{L_o} > i$ then for any $z \leq L_o$, $\theta(z, t)$ will be given as in (5), since the wetting front for that particular location (corresponding to $K_s$) will have moved further downward than $L_o$. Likewise, for any value of $i < K_s < K_{L_o}$ for depths $z \geq L_o$: $\theta_t = \theta_i$, since the wetting front corresponding to that $K_s$ value/location will not yet have reached $z = L_o$. If $K_s(\mathbf{x}) < \mathbf{i}$ and $t - t_o$ is greater than the time to ponding, then the grid cell containing $\mathbf{x}$ is saturated, and the




relationship between $K_s$ and $L_f$ is determined by the Green and Ampt model as in Charbeneau (2000):

$$K_s = \frac{1}{t - t_p}\left[L_f(\theta_s - \theta_i) - F_p + (H + \Psi_f)(\theta_s - \theta_i)\ln\left(\frac{H + \Psi_f(\theta_s - \theta_i) + F_p}{H + \Psi_f(\theta_s - \theta_i) + L_f(\theta_s - \theta_i)}\right)\right] \quad \text{if} \quad t > t_p, \tag{9}$$

$$F(t) = it \quad \text{if} \quad t < t_p. \tag{10}$$

In the above, $H$ is the ponding depth above the ground surface, $\Psi_f$ is the constant effective suction head at the wetting front, and $F_p$ is the water that has infiltrated prior to ponding. Assuming all water infiltrates into the soil prior to ponding when $i > K_s$, then $F_p = it_p$.

In the case in which $K_s < i$, for a particular $L = L_o$, $K_{L_o}$ is found using (9) with $L_f$ set to $L_o$. If $i > K_s > K_{L_o}$ then for all $z < L_o$, $s(z, t; K_s) = 1$. If $K_s < K_{L_o}$ then $s(z = L_o, t; K_s) = s_i$.

The probabilistic properties of the saturation fields are constructed using Monte Carlo simulation since analytic reconstruction of the finite dimensional probability distributions is difficult. During the phase in which $K_s(\mathbf{x}_o) < i$ and the Green and Ampt model (9) and (10) are used, analytical determination of the density of $s(\mathbf{x}_o, t; K_s)$ is difficult because there is no known expression in terms of simple rectangular profiles known of the behavior prior to ponding. In this case the depth to the wetting front is determined by conservation and the assumption that all the water infiltrates into the soil prior to surface ponding.

Using this set of conditions, the minimum wetting front depth $L_m$, and some imposed maximum depth $L_M$, a partition can be formed of $[L_0, L_M] : \{L_0, L_1, L_2, \ldots, L_{M-1}, L_M\}$. For each member of the partition $L_j$, the corresponding $K_{L_j}$ may be found and the values of the $K_s$ in each grid cell will indicate the value of the saturation $s(z = L_j, t; K_s)$ or equivalently the water content. This is done for each time in the partition of time values up to the cessation of rainfall, $t_r$.

An alternative is to simply solve in each grid cell for $L_f$ using the value of $K_s$ assigned to the cell. This is simple when $K_s > i$ if (6) is used, and when $K_s < i$, an iterative method (e.g. Newton-Raphson) must be used to solve for $L_f$ in (9). The time to ponding can be found at each grid cell for each realization of $K_s$ through (7). This technique requires a more sophisticated implementation, though it may be preferable if precise values of the wetting front depths at each time step are wanted.

The dichotomy between the saturated and unsaturated regions induces the concern of the probability that a zone falls into the unsaturated regime. For rainfall rate held constant, this will depend on the hydraulic conductivity random variable and the time at which the measurement is taken. A cell will be unsaturated if $K_s > i$. The soil matrix in a cell will be at saturation if the rainfall rate exceeds the $K_s$ and $t > t_p$:

$$K_s < \frac{i^2 t}{it + \Psi_f(\theta_s - \theta_i)} = i\left(\frac{1}{1 + \frac{\Psi_f}{it}(\theta_s - \theta_i)}\right) \leq i. \tag{11}$$

Lastly, a cell can be characterized as having the rainfall rate exceeding the saturated hydraulic conductivity, but at pre-ponding:

$$i\left(\frac{1}{1 + \frac{\Psi_f}{it}(\theta_s - \theta_i)}\right) \leq K_s \leq i. \tag{12}$$

The relationships derived above allow calculation of probabilities for the finite dimensional distributions of $K_s$ for the occurrence of saturation with ponding, infiltration beyond the maximum value of $K_s$ but pre-ponded, and infiltration below





the maximum value of the saturated hydraulic conductivity. As mentioned above, it is hard to characterize the sets (in terms of $K_s$, $i$, and $t$) in which $\theta(t; K_s) = \theta_o$ for $\theta_r \le \theta_o \le \theta_s$ when there are cells in which $i > K_s$ and $t < t_p(K_s)$. This is because the behavior of $\theta(t; K_s)$ for values (12) is unknown in terms of a rectangular profile approximation. In the event that $t > t_p(K_s)$ in all cells in which $i > K_s$, the stochastic properties of $s(\mathbf{x}, t; K_s)$ can be found analytically. Notice that:

$$\lim_{t \to t_p} i \left( \frac{1}{1 + \frac{\psi}{it}(\theta_s - \theta_i)} \right) = K_s.$$

Thus the inequalities (11) and (12) collapse down to $K_s \le i$.

The variable saturation rectangular soil water content profile model in Chen et al. (1994a, b) was able to characterize the saturation in terms of values of $K_s$ using a different approach. The technique presented here is simpler to apply and makes use of the widely accepted Green and Ampt infiltration model (9), (10). However, since there is no known representation for $s(x, t; K_s)$ for $K_s$ values (12), any complete representation of the saturation field is impossible while there are pre-ponded cells in which $i > K_s$. After a certain amount of time has passed so that $t > \max_{K_s} t_p(K_s)$, a complete representation of the random field $s(x, t; K_s)$ will become available. Either analytical or approximate finite dimensional distribution functions may be used depending on the complexity of the field $K_s(\mathbf{x})$.

## 4 Solutions based on Stochastic Interpolation of Parameters

In this solution approach $K_s$ is allowed to vary as a random function of space and time. In engineering applications, it is not often the case that $K_s$ values are known at all points in a grid. In sparsely gauged watersheds, it is necessary to interpolate between measured points. A stochastic process is generally used to assign values to unknown grid cells for the saturated hydraulic conductivity.

In general, different processes are used for each variable. It is popular to use Gaussian processes to represent $K_s$, e.g. a stationary lognormal multivariate random variable, or a non stationary geometric fractional Brownian field. The fractional Brownian fields were generated here using (Kroese and Botev, 2015, Chapter 12).

Once the proper processes are in place for the hydraulic conductivity, then the stochastic water content and wetting front depths for each grid cell can be determined. It is usual in applications to determine the statistical properties of the state variables, e.g. the mean, variance, and covariance. The multivariate distribution function for the state variables over the grid points at fixed time instances can be approximated by forming a multivariate histogram over a limited number of realizations of the variables. The realizations are simulated by computing the state variables with fixed generated values for the parameters gained through realizations of the stochastic processes used to model them. Similarly, the marginal probability density functions can be realized by constructing the univariate histogram at a single point in space-time. Evolution in time can be seen through a time series analysis of either the statistical properties or the histograms.

A second option for deriving the time-space evolution of the probability density functions of the state variables is available through the general evolution equations for hydrologic processes given in Kavvas (2003). A continuity equation for the density of hydrologic state variables is derived in the phase space, and the ensemble average is taken to produce an equation for the




evolution of the probability density function for the state variables. An exact closure to the order of the covariance is derived. The probabilistic evolution equation takes the form of a Fokker-Plank equation, but it is a *deterministic* equation for the probability density function. The advantage of this method is that it allows determination of the probabilistic evolution without the need for a large amount of Monte Carlo trials to determine either the statistical properties, or any particular distributions

for the multivariate or marginal processes. This may be extremely useful for large domains or high resolution models in which generating the hydraulic conductivity and rainfall random fields may be expensive.

The modeling strategy here relies on the standard Green and Ampt model to determine solutions when the rainfall rate exceeds the theoretical maximum saturated hydraulic conductivity. In the event that $i > K_s(x_o)$ but surface ponding has not yet occurred, the water content is unknown for the grid cell containing $x_o$. The soil water content for all grid cells in the field

will only be known when all cells in which $i > K_s(x_o)$ are at saturation (i.e. surface ponding has occurred). It is of interest to know for a given hydraulic conductivity random field $K_s(\mathbf{x})$ at a specified time $t_o$, what are the finite dimensional probabilities that for a set of grid cells, either $i > K_s$ and $t > t_p$ or $i < K_s$. This event corresponds to the existence of the finite dimensional distributions of the water content in soil under the variably saturated rectangular soil water content profile model. From the discussion above, this can be simply determined from (11) and (12):

$$\Pr(\text{no pre-ponding}) = 1 - \Pr\left[ i \left( \frac{1}{1 + \frac{\psi}{it}(\theta_s - \theta_i)} \right) \leq K_s \leq i \right]. \tag{13}$$

The probability that a grid location is unsaturated (and not pre-ponded) can be determined directly from the marginal distribution of the spatial process $K_s$. This can be expressed as: $\Pr(S_t(\mathbf{x}_o) < 1) = \Pr(K_s(\mathbf{x}_o) > i(\mathbf{x}_o, t))$. For certain $K_s$ processes the probabilities can be easily determined. For example, if $K_s$ has a stationary lognormal distribution then the multidimensional probabilities are just the products of the 1D marginal probabilities. The same is true if $K_s$ is modeled by Brownian motion. The

fractional Brownian processes have known finite dimensional characteristic functions; in those cases the multivariate probabilities can be determined through the Fourier transform of the characteristic function. The finite dimensional probabilities can be used to analytically determine mean run times until ponding is reached for sets of grid cells.

In numerical simulation, Monte Carlo simulation particularly, a finite number of realizations will be generated. The maximum time to ponding over all realizations of the $K_s$ field can be established through (7). The distributions and statistical

moments of the hydrologic processes are developed after that maximum ponding time. The expected ponding time can be used to judge applicability of the model *a priori*. If the expected $t_p$ is greater than the expected or target duration of the wetting event, then the model may be inappropriate.

## 5   Numerical Simulations

Simulations to explore this new approach were performed for a heterogeneous field with $K_s$ simulated through a fraction

Brownian process and rainfall rate kept at a constant value. The simulations only extended up through the wetting phase. MATLAB was used to generate the $K_s$ random fields and to carry out the implementation of the rectangular profile approximation. The goal of these demonstrations was to develop solutions for the statistical and probabilistic properties of the water content





and the depth to the wetting front. 500 trials were performed with independently generated log $K_s$ fractional Brownian fields. For each realization, the soil water content and depth to the wetting front were found as a function of the particular $K_s$ field realization. The soil characteristics for all experiments are consistent with a sandy type soil. Specifically, $\theta_s = 0.437$, $\theta_o = 0.02$, $\theta_i = 0.2001$, $l = 1.0$, $\eta = 0.592$, and $|\Psi_f| = 4.96$ cm.

The hydraulic conductivity was generated through a fractional Brownian field simulation with the Hurst coefficients $H = \{0.3, 0.5, 0.7\}$. The rainfall rate was set at $i = 0.0111$ mm/s (40 mm/hr) with a duration of 10 hours (36000 sec). For each set of simulations, a circular grid with a uniform horizontal increment of $dx = dy = 1$mm was generated for the $K_s$ fields. A circular grid was taken so the covariance function for the random field, defined as $\mathrm{Cov}(X_{\mathbf{t}}, X_{\mathbf{s}}) = \|\mathbf{t}\|^{2H} + \|\mathbf{s}\|^{2H} - \|\mathbf{t} - \mathbf{s}\|^{2H}$ is symmetric in the domain. The hydraulic conductivity fields were generated to correspond with more permeable soils with

averaged values between 0.03 and 0.06 mm/s. Stochastic processes were chosen so that both cases $K_s > i$ and $K_s < i$ would have an appreciable representation within the field for the ensemble of realizations.

    The averaged wetting front depths at the end of the simulation are displayed in figures 2, 3, and 4. The depth to the wetting front is in mm below the soil surface. Figures 5, 6, 7 show the averaged saturation at the end of the simulation in mm$^3$/mm$^3$. The averaged saturated hydraulic conductivity is presented in figures 8, 9, 10.

To illustrate the application of the model, a series of single run simulations were done to demonstrate the model results for two cases of saturated hydraulic conductivity fields generated from exponentiated fBm (H = 0.3,0.5). The holes in the initial saturation fields are grid cells where $i > K_s$. Initially $t < t_p$ in any cell where the rainfall intensity exceed $K_s$. By the end of the simulation, most cells will have $t = t_r > t_p$, but in some cases $t_p > t_r$. When the latter case occurs the saturation field will have a hole as well. This highlights the need for a representation for $\theta(t)$ when $t < t_p$. The results of the one-run simulations

at the first time step can be seen in figures 13 and 15. The profiles at the end of the simulation can be seen in figures 14 and 16. The hydraulic conductivity fields for the single run simulations are displayed in figure 11 and 12.

## 6   Conclusions

The variable saturated model for vegetated regions is a useful tool for modeling infiltration into the soil unsaturated zone for wetting events in which the rainfall rate or infiltration rate is smaller than the saturated hydraulic conductivity. In these

situations, ponding at the surface should not be expected and there are few tools for modeling these cases. This method can be joined with a saturated rectangular soil water content profile model to provide a complete model for the wetting of a heterogeneous field subject to variable rainfall in both space and time. In actual wetting events there will be patches that reach saturation and patches that never reach saturation, particularly in soils and rainfall events where the magnitudes of the rainfall rate and the hydraulic conductance of the soil have large variances. This method allows for both instances and is thus well

suited to model realistic infiltration into a heterogeneous soil during rainfall events.

    It was observed in the course of numerical simulation of the non linear Richards' equation, displayed in Figure 1, that the hydraulic conductivity either quickly or instantly (e.g. in the first time step) equilibrates with the infiltration rate in the case when $i < K_s$. The approximation tends to underestimate the actual solution.





This method is simple to apply and can easily be extended to account for additional phenomena, such as redistribution and evapotranspiration. Much like the other rectangular profile approximations, a solution for each realization of a subsurface problem modeled with a random hydraulic conductivity field can be found without excessive computational effort. The ensemble

5  of the head or water content fields at any point in space and time can be used either to approximate stochastic distributions, or form a field of averaged values and accompanying variance/covariance fields. Solutions gained through this approach are also amenable to the theoretical results developed in Kavvas (1999, 2003) for determining the probablistic behaviour for the time/space evolution of the water content or head random functions analytically to the order of the covariance.





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





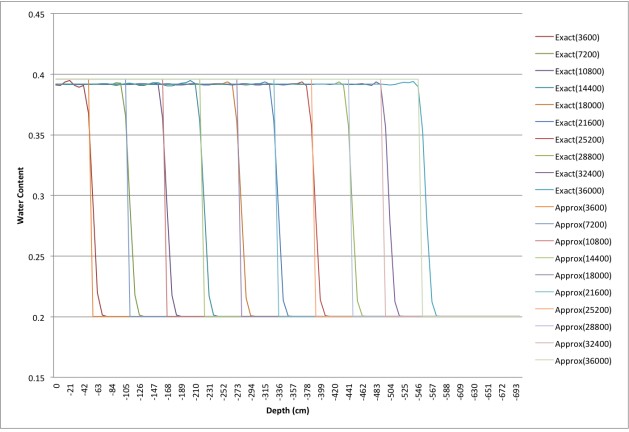

**Figure 1.** Superimposed solutions of Richard's equation from HYDRUS 1D and approximate solutions gained through the new variably saturated model for times taken at 3,600 second intervals.

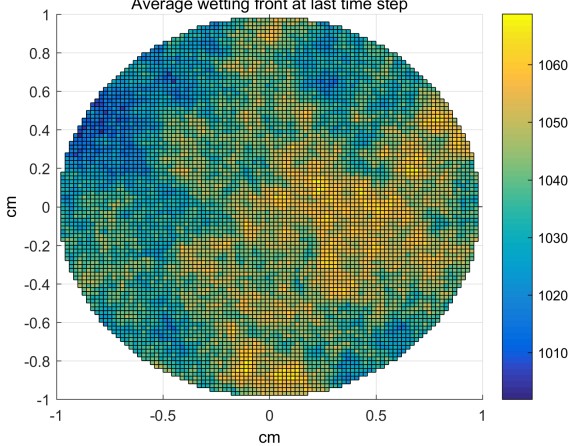

**Figure 2.** Averaged wetting front depths [mm] for simulations with Hurst coefficient $H = 0.3$





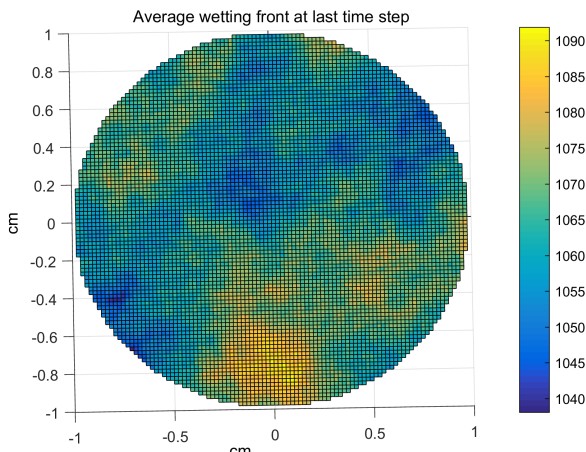

**Figure 3.** Averaged wetting front depths [mm] for simulations with Hurst coefficient $H = 0.5$

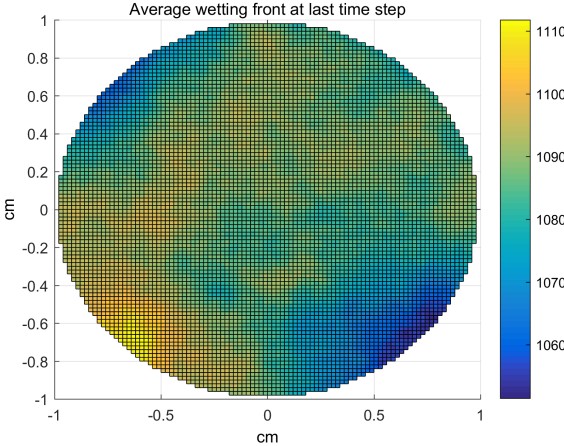

**Figure 4.** Averaged wetting front depths [mm] for simulations with Hurst coefficient $H = 0.7$




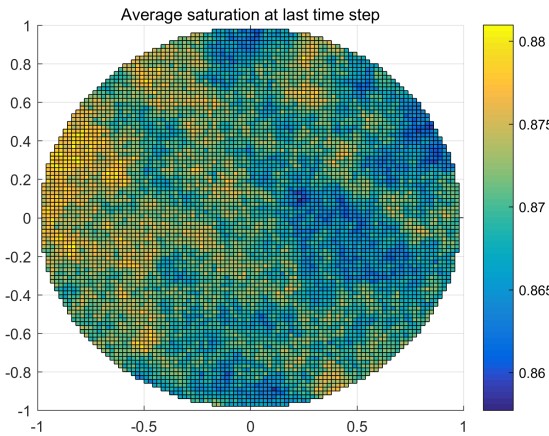

**Figure 5.** Averaged saturation for simulations with Hurst coefficient $H = 0.3$.

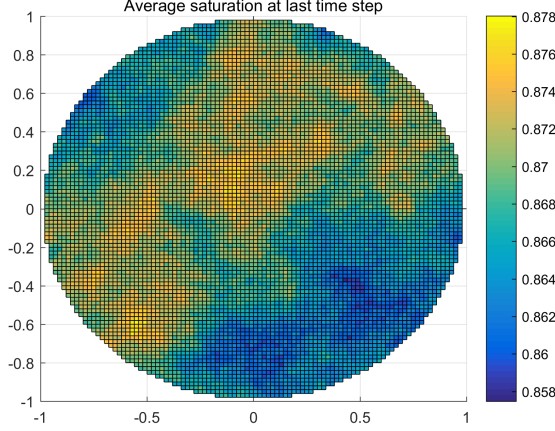

**Figure 6.** Averaged saturation for simulations with Hurst coefficient $H = 0.5$.





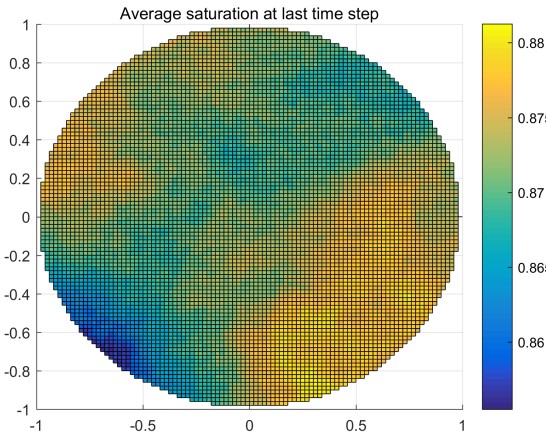

**Figure 7.** Averaged saturation for simulations with Hurst coefficient $H = 0.7$.

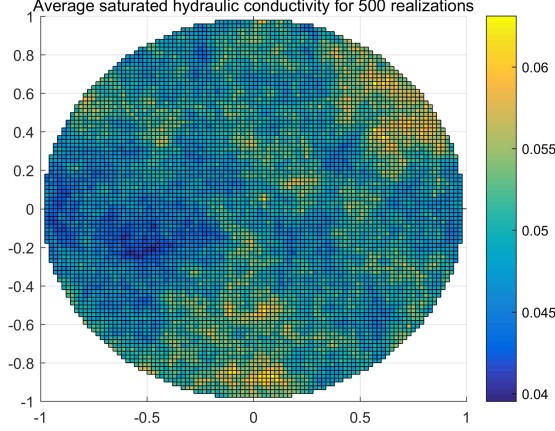

**Figure 8.** Averaged saturated hydraulic conductivity fields [cm/hr] for fBm conductivity fields with $H = 0.3$.




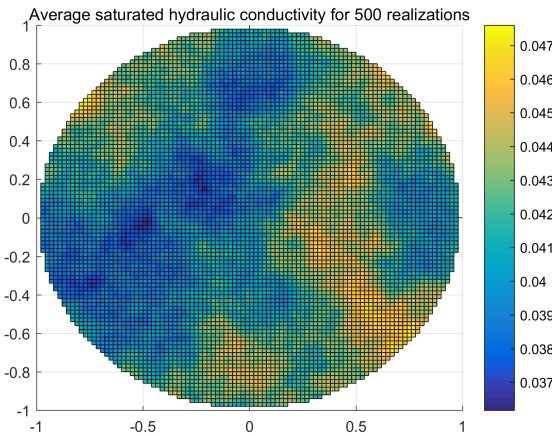

**Figure 9.** Averaged saturated hydraulic conductivity fields [cm/hr] for fBm conductivity fields with $H = 0.5$.

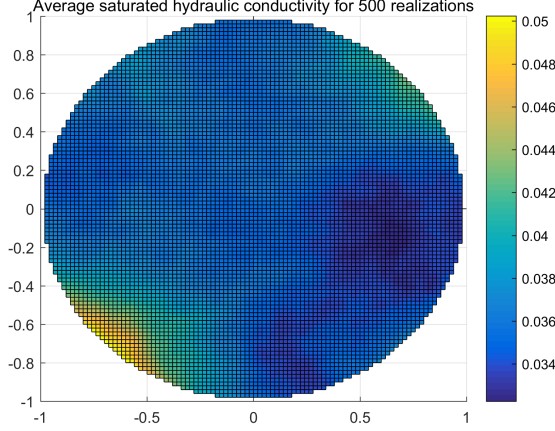

**Figure 10.** Averaged saturated hydraulic conductivity fields [cm/hr] for fBm conductivity fields with $H = 0.7$.





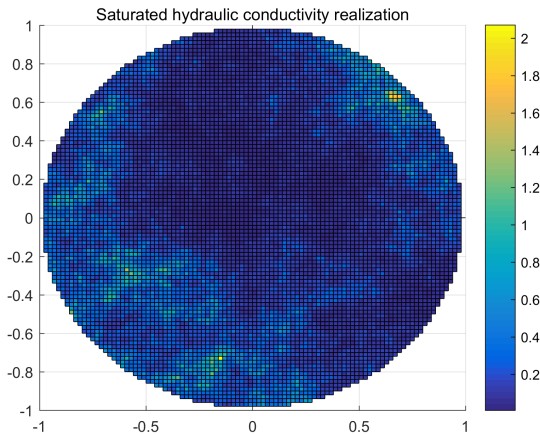

**Figure 11.** $K_s$ realization from a fractional Brownian field with Hurst coefficient $H = 0.3$.

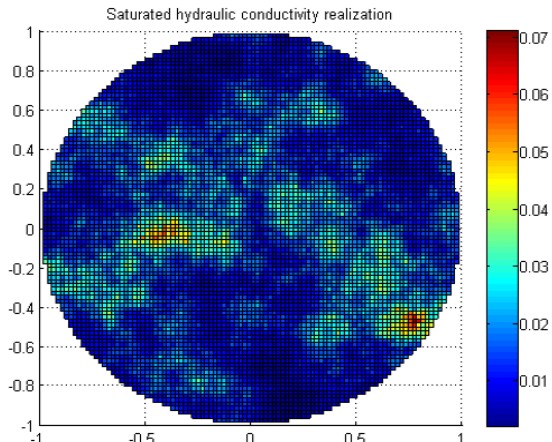

**Figure 12.** $K_s$ realization from a fractional Brownian field with Hurst coefficient $H = 0.5$.





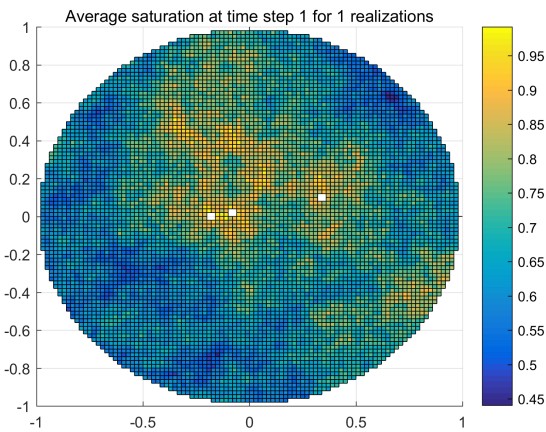

**Figure 13.** Example of a single run of VSVI model with $H = 0.3$ and $i = 0.0111$ mm/s. Initial saturation profile.

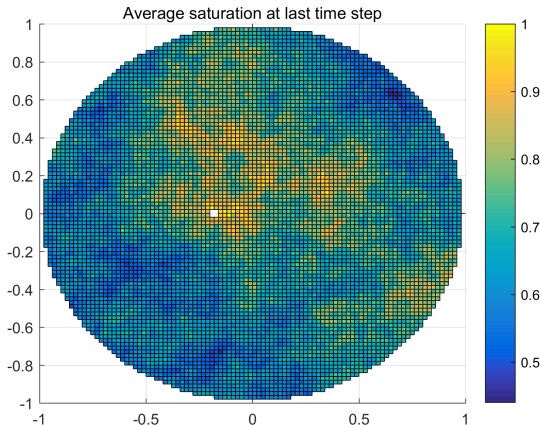

**Figure 14.** Example of a single run of VSVI model with $H = 0.3$ and $i = 0.0111$ mm/s. Saturation profile at the end of simulation.





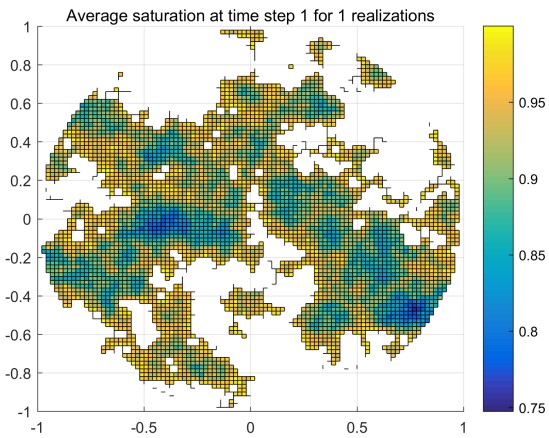

**Figure 15.** Example of a single run of VSVI model with $H = 0.5$ and $i = 0.0111$ mm/s. Initial saturation profile.

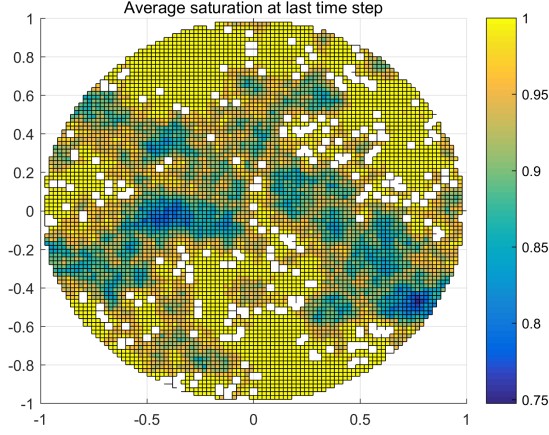

**Figure 16.** Example of a single run of VSVI model with $H = 0.5$ and $i = 0.0111$ mm/s. Saturation profile at the end of simulation.