# Peer review of "Variable Saturation Infiltration Model for Highly Vegetated Regions"

_Hydrology and Earth System Sciences, 2016_

## Referee Comment (RC1) · Anonymous Referee #1 · 8 Apr 2016

This manuscript presents a newly-developed hydrological model based on simple analytical solution, which is integrated with a probability distribution function to quantify the impacts of distributed saturated hydraulic conductivity ($K_s$) on soil moisture dynamics during the rain-event. The hydrological model considers the wetting front in the unsaturated soil as a piston-shape (similar to the concept of Kinematic Wave approximation for unsaturated flow), while the wetting front below the ponding soil surface is calculated with the Green-Ampt equation. The fractional Brownian field is used for parameterizing the saturated hydraulic conductivity ($K_s$) to account the soil heterogeneity. With the proposed model, a group of numerical tests is provided attempting to simulated soil moisture distribution before and after rainfall under different Hurst coefficients of the fractional Brownian field.

The paper is indeed within the scope of an important and interesting research topic

of quantifying the impact of soil heterogeneity on soil moisture distribution during rain-event. However, the authors did not clearly clarify the innovative insight and scientific findings from their numerical experiments. Besides, the content of the manuscript is not well-organized, and the manuscript needs significant revision to meet the requirement of publishing in an international journal.

Comments and suggestions related to each specific section:

In the abstract the authors discussed the disadvantage of rectangular profile method and Darcy-Richards equation, which is not the reason why the new model needs to be developed as many of existing hydrological models use analytical solution to describe the variably-saturated flow. The structure of the abstract needs to be refined, because after reading it the reader might still not be able to know what is the focus of this study.

The introduction lacks of a convincing clarification of the necessity to conduct this study. There are two paragraphs (Page2, Line 12-35) that criticize the assumptions and limitations of Richards' equation and Green-Ampt equation, while those knowledge are well-acknowledged and can be found in most of textbook in the field of hydrology. I would expect to see a more systematic review of how to represent the soil heterogeneity in hydrological modeling system. Most importantly, the authors should discuss the impact of using different distribution functions of $K_s$ on the patterns of soil moisture dynamics.

The content of method in section 3.1 is not well-organized. The authors provide a benchmark study by comparing the solution of piston-shape wetting-front equation with the numerical solution of HYDRUS-1D, but it is already known that the solution of Richards' equation is quite similar to a piston shape for coarse soil texture. Moreover, the benchmark case only accounts the sandy soil, while the authors do not mention that the discrepancy between the simulation results of these two models will be much larger when it comes to fine-texture soil.

In section 3.2, Line 20, there is a confusing statement of "The transition may be accomplished through a linear model, such as rectangular profile, or through non linear dynamical equations such as Richards' equation, or the kinematic wave equation". The authors should either explicitly indicate what is the method adopted in your model to calculate the "transition back to the variably saturated profile" and provide numerical example, or clearly indicate that you did not consider the redistribution of soil moisture after rain-event.

In section 3.3, the assumptions of the proposed model should be more clearly clarified. If I am correct, in each computational block, the initial water content is a constant value along the vertical soil profile (uniform distribution), and similarly the $K_s$ is also a constant value along the vertical direction. Therefore, the proposed model only accounts the soil heterogeneity at horizontal direction but not the vertical direction. At least, the authors should clearly state that soil heterogeneity at vertical direction is not accounted if I am correct. Besides, I did not find that the authors discuss why to use those assumptions.

Section 4, the authors should clarity why the Brownian field distribution is used rather than the Gaussian distribution. Besides, it is might necessary to clearly discuss why the Brownian field distribution can be more realist if there are some the field evidences to support your study. The content of this section is mainly describing the mathematical aspect of the fractional Brownian field, while I believe that the readers of HESS also eager to know the physical meaning of the formulation that you used.

The content of Section 5 is not clear in terms of initial condition setting and results description. The authors did not clarify what new insight we can get from the numerical simulations. The grid size of 1 mm (Page 10, Line 7) is small for a hydrological system. The author should clarify why such fine grid is used and what we can conclude from your numerical simulation.

In the conclusion section, the author state that the proposed approach can be easily "extended to account for additional phenomena". However, the current study only

show the result of the soil moisture dynamics in sandy soil under very simple boundary condition (single rain event with constant rainfall intensity) and initial condition (initial soil moisture along the vertical profile might be a constant value). Those assumptions imply the numerical results are based on drastically simplified condition, which does not guarantee the propose approach can be used for more complex case study.

The results only show the moisture dynamics in sandy soil under simple condition. It is also favorable if authors can provide more systematic numerical results, such as using your model to calculate the soil moisture dynamics under more complex rainfall events, or more complex initial and boundary conditions in different soil types.

There are number of technical errors should be corrected, below gives a few examples:

Some of the terminologies used in this study is heavy-going and inconsistent. I guess the "the rate of hydraulic conductance" (Page 1 Line 14) means hydraulic conductivity, and the "ability of soil to absorb moisture" (Page 1, Line 20) means infiltration capacity. I would suggest that the manuscript should use widely-used terminologies, or the author should explain the special meanings of those uncommon terminologies.

The numbering of equation is incorrect: the first equation on page 4 was not labeled.

The author should clearly introduce the symbol of variable that used in this paper, for example, $\phi$ in Eq.(1), $V$ in Eq.(20), and parameters of Brooks-Corey model.

The authors should consistently use academic writing style to write the manuscript and to avoid the technical mistakes that suggested above.

---

## Referee Comment (RC2) · Anonymous Referee #2 · 24 Apr 2016

This manuscript describes an infiltration model in 1 dimensional column that is similar to the Green-Ampt approximation. Such a practical deterministic model for a 1 dimensional uniform porous media for an equilibrium condition may not be new. Then, the Monte-Carlo simulation was used to analyze the horizontal hydraulic conductivity variability. I would classify this model as another extended/modified GA model, rather than a new water infiltration model. This study neglected the vertical soil variability while it is highly arguable. The assumptions used in this work should be clearly stated in the article. Meanwhile, it may be an interesting opinion that the stochasticity of the rainfall pattern essentially controls the heterogeneity of the infiltration in highly permeable watershed. This manuscript did not effectively emphasize this opinion in the argument. In general, the explanations or justifications for the proposed methodology seem to be weak. For example, no justification for the stochastic modeling (fractional Browning

modeling) was provided. The parameters (for example, Hurst number = 0.3, 0.5, 0.7) were not justified either. There is no validation with any field data. Only one model comparison against the numerical solution of 1D Richards equation was provided under one simplistic condition with a constant water supply. This may be the weakest point of this manuscript. Also, the title indicates the effect of vegetation while no plant water intake component was discussed at all. The authors may consider either changing the title or adding the vegetation component. Format and readability is below expectation as a publication ready manuscript. The coherence of the text and figures should be improved. Therefore, I would recommend major revisions for this manuscript.

---

## Author Comment (AC1) · 16 Jun 2016

Response to Referee #1:

Generally, I will make necessary clarifications and re-organize the article in order to give the best presentation of the ideas in this article.

Response to comments on the abstract: I will rewrite the abstract to introduce the Variably Saturated Rectangular Profile model, comment briefly and clearly on what situations it may be useful, and the model's computational efficiencies. The focus of the article should be clear to every reader after the abstract, and I will fix this section to ensure that.

Response to comments on the introduction: I agree that the limitations of Richards

equations and the Green and Ampt model are well documented; therefore these discussions may be shortened or removed. One of the primary advantages of this model is it's ability to model flows in horizontally heterogeneous soils. I agree that a discussion of hydraulic conductivity distribution on soil moisture dynamics will be interesting. A systematic review of representations of hydraulic conductivity could be useful given the utility of this particular model.

Response to comments on section 3.1: I will re-organize this section to explain the methodology of the sub-saturated model and can investigate comparisons with Richards equation for soil hydraulic conductivities corresponding to finer grained soils.

Response to comments on section 3.2: I will make it clear that I am not addressing the redistribution phase of soil water.

Response to comments on section 3.3: I will clarify that horizontal heterogeneity alone is considered and state explicitly the assumptions on the model.

Response to comments on section 4: There have been both field studies and theoretical studies on the merits of Brownian and fractional Brownian fields as models for heterogeneous soils. I will provide proper references for these studies.

Response to comments on section 5: The initial condition for the numerical simulations will be determined by the initial saturated hydraulic conductivity (which is constant in time in these trials), and the initial rainfall rate (also constant in these trials). If the former is larger than the latter the initial condition of the moisture can be determined, if the latter is larger than the former then the initial moisture content cannot be determined under the assumptions of the Green & Ampt model. The moisture content will be the saturated content after the time to ponding has been reached. This will be clarified and expanded upon in my revisions. The grid size is small for a typical hydrologic study, it was picked as a matter of convenience in generating the Ks fields. However, since Brownian and fractional Brownian fields are self-similar, the grid and the Ks fields can be scaled to a more hydrologically relevant size.

Response to comments on the conclusion: The goal of the numerical demonstrations was to show the dynamics of the model under degrees of variability in the soil type (the soil Ks). The Ks fields were picked to have fields with differing soils (or differing vegetation) where the magnitude of the Ks changed by an order of magnitude or more. See Figure 6, there are instances of three fields with different Hurst coefficients, with values of Ks at different cells varying between less than 0.1 and 2 (H = 0.3), less than 0.01 and 0.07 (H = 0.5), and less than 0.01 and more than 0.07 (H = 0.7). The noisiness of the three fields varied greatly, with H = 0.3 anti-persistent, H = 0.7 persistent, and H = 0.5 classical Brownian motion. The ease of modeling such irregular fields was the focus of the experiments. More complicated rainfall conditions could be attempted, but a storm event with multiple rainfall events would involve significant complication due to the redistribution (not a focus of the study) and would not truly add dimension. A single variable rainfall event will be included in the revision. Numerical simulations with at least two different soil texture classes will be included and the different soil textures clearly identified.

I will correct all the technical errors and inconsistent terminology that were pointed out.
* * *

---

## Author Comment (AC2) · 16 Jun 2016

Response to Referee #2:

This model resembles the Green and Ampt model in the respect that it approximates the movement of water in a piston-like fashion. However, it differs and is applicable to situations where the Green and Ampt model cannot be used; any situation in which water does not pond on the surface, or does not saturate the soil. Certainly this situation occurs widely in both natural and engineered environments. That being said, in certain respects this model is not new: as referred to in the article, Chen et al. developed a variable saturation rectangular profile model in two 1994 publications. This model differs from the 1994 Chen model in its assumption and in that it is very accessible; indeed, its application is very similar to, and in some ways simpler than, the Green and

[Figure]

Interactive
comment

Ampt model. The novelty of the model is much more in its applicability than anything else. The simplicity of the model enables a modeler to apply it to situation in which the variability in the soil is extreme. Though I have only looked at horizontal heterogeneity in the soil, the model could be examined and extended by others to include vertical heterogeneity. Furthermore, the model has been presented to include both variability in the rainfall rates and the soil hydraulic conductivity. Although I did not include variable rainfall rates in the numerical simulations, they could be included. A numerical simulation with a single variable rainfall event will be included.

I agree that the idea that the variability in the rainfall rate controls the heterogeneity of the soil is interesting. The type of soil and factors that may increase the hydraulic conductivity of the soil must be considered along with stochasticity in the rainfall pattern; this is why I referred specifically to highly vegetated soils in the title and at parts of the article.

I neglected both variable rainfall patterns and vertical heterogeneity, both of which are extremely important in applications, because I did not want to couple the process to redistribution of soil moisture. This process could be incorporated in the same way as it is in Green and Ampt. I will make it clear that the redistribution phase of soil moisture is not considered in this paper.

The criticism that the stochastic model for Ks was not justified or commented on is valid. I chose fractional Brownian motion principally because of the inspirational works of Mandelbrot, Wallis, and Van Ness among others. Field experiments using fBM and similar stochastic processes have been done by researchers such as S. Painter. I will give the proper references and comments directing readers to these works in my revision. I picked the Hurst numbers 0.3 and 0.7 to correspond to anti-persistent and persistent random field respectively. The anti-persistent case produced extremely noisy and differences of several orders of magnitude in the Ks values of the soil cells. The persistent case produced large almost contiguous areas of similar Ks values (typically far from the average Ks value). The H = 0.5 case was standard Brownian motion has

more familiar Gaussian behavior. I thought all of these cases may be of interest to researchers in representing highly heterogeneous soils. These cases are discussed by the authors listed above as well. The method in which the variability of Ks is represented makes little difference in the application of the method however. One could just as easily model Ks using a more common cell-wise-independent Log-Normal field if desired. I will make these statements in the revised paper and clearly explain why I have focused on fractional Brownian fields.

The method was compared to a relatively simple application of Richards equation for reasons similar to those listed, and due to the complications involved in simulation a highly horizontally heterogeneous aquifer with Richards equation. I will include simulations for finer grained soils.

I will reconsider the format, readability, and coherence of the Figures and the text in order to clarify the points discussed above.